# Comprehensive Evaluation of Immune-Checkpoint DNA Cancer Vaccines in a Rat Cholangiocarcinoma Model

**DOI:** 10.3390/vaccines8040703

**Published:** 2020-11-24

**Authors:** Yi-Ru Pan, Chiao-En Wu, Ming-Huang Chen, Wen-Kuan Huang, Hsuan-Jen Shih, Keng-Li Lan, Chun-Nan Yeh

**Affiliations:** 1Liver Research Center, Department of General Surgery, Chang Gung Memorial Hospital, Linkou Branch, Chang Gung University, Taoyuan 333, Taiwan; panyiru0331@gmail.com; 2Division of Hematology-Oncology, Department of Internal Medicine, Chang Gung Memorial Hospital, Linkou, Chang Gung University College of Medicine, Taoyuan 333, Taiwan; jiaoen@gmail.com (C.-E.W.); medfoxtaiwan@gmail.com (W.-K.H.); samuelshih@cgmh.org.tw (H.-J.S.); 3Center for Immuno-Oncology, Department of Oncology, Taipei Veterans General Hospital, Taipei 112, Taiwan; mhchen9@vghtpe.gov.tw; 4School of Medicine, National Yang-Ming University, Taipei 112, Taiwan; 5Cancer Center Karolinska, Department of Oncology-Pathology, Karolinska University Hospital, 17176 Stockholm, Sweden; 6Department of Oncology, Taipei Veterans General Hospital, Taipei 112, Taiwan; 7Institute of Traditional Medicine, School of Medicine, National Yang-Ming University, Taipei 112, Taiwan

**Keywords:** cholangiocarcinoma, PD-1, PD-L1, CTLA4, DNA vaccine

## Abstract

Cholangiocarcinoma (CCA) is a malignant tumor with aggressive biological behavior. Immune checkpoints such as cytotoxic T-lymphocyte antigen 4 (CTLA4) and antiprogrammed death 1 (PD-1) are critical immune-checkpoint molecules that repress T-cell activation. The DNA vaccine potential against CTLA4 and PD-1 in CCA is unknown. We used a thioacetamide (TAA)-induced intrahepatic cholangiocarcinoma (iCCA) rat model to investigate the DNA vaccine potential against CTLA4, PD-1, and PD-L1. We detected PD-L1 expression in CCA and CD8^+^ T-cell infiltration during CCA progression in rats. We validated antibody production, carcinogenesis, and CD8^+^ T-cell infiltration in rats receiving DNA vaccination against PD-1, PD-L1, or CTLA4. In our TAA-induced iCCA rat model, the expression of PD-L1 and the infiltration of CD8^+^ T cells increased as in rat CCA tumorigenesis. PD-1 antibodies in rats were not increased after receiving PD-1 DNA vaccination, and CCA tumor growth was not suppressed. However, in rats receiving PD-L1–CTLA4 DNA vaccination, CCA tumor growth was inhibited, and the antibodies of PD-L1 and CTLA4 were produced. Furthermore, the number of CD8^+^ T cells was enhanced after PD-L1–CTLA4 DNA vaccination. DNA vaccination targeting CTLA4–PD-L1 triggered the production of specific antibodies and suppressed tumor growth in TAA-induced iCCA rats.

## 1. Introduction

Immune checkpoints such as cytotoxic T-lymphocyte antigen 4 (CTLA4) [1] and antiprogrammed death 1 (PD-1) [2], discovered by James P. Allison and Tasuku Honjo, respectively, were found to block antitumor immunity. Monoclonal antibodies targeting immune checkpoints were designed to block the negative interactions between cancer and immune cells, thereby enhancing tumor immunity. Nowadays, immune-checkpoint inhibitors (ICIs) are widely used in the treatment of various cancers such as melanoma [3,4], nonsmall-cell lung cancer [5,6,7], small-cell lung cancer [8,9,10,11], and urothelial carcinoma [12].

Intrahepatic cholangiocarcinoma (iCCA) arising from the epithelium of bile ducts is a rare but aggressive malignancy [13,14,15]. Most patients with iCCA are diagnosed at an advanced stage and have extremely poor prognosis. Since 2010, chemotherapy with gemcitabine and cisplatin has been the mainstay of treatment [16], and no targeted or novel therapy has been approved on the basis of Phase III studies [17,18,19]. ICIs showed modest efficacy in the treatment of iCCA [20]; currently, pembrolizumab is merely indicated for iCCA patients with microsatellite instability-high (MSI-H) or deficient mismatch repair (dMMR) [21].

In previous studies [22,23], thioacetamide (TAA)-induced iCCAs in rats were compared with human iCCAs. In histologic features, multifocal bile ductular proliferation, histologic atypia, and invasive intestinal-type CCA were chronologically determined, and similar expression patterns of c-Met and c-ErbB-2, EGFR, apomucins, and MMPs (Matrix metallopeptidases) were detected in rat TAA-induced iCCAs and human CCAs, suggesting that the progression of TAA-induced iCCA can mimic the multistep model of human CCA.

Therapeutic DNA cancer vaccines are considered a promising strategy to activate the immune system, resulting in tumor inhibition [24,25,26]. In previous early-phase studies, DNA vaccines were designed to target tumor antigens such as mutational antigens or tumor-associated antigens rather than immune checkpoints [24,25]. However, such therapeutic DNA cancer vaccines only demonstrated limited efficacy in early clinical trials, with an acceptable safety profile possibly resulting from the immunosuppressive mechanisms developed by the tumor [24]. Moreover, increasing the immunogenicity of DNA vaccines and combining with other therapies (such as ICIs) may improve their activity by enhancing immune-cell activity or suppressing immunosuppression in the tumor microenvironment [24,27].

To further investigate the immune-checkpoint blockade in iCCA, this study aims to use rat TAA-induced iCCA as an experimental model to analyze the immune response in iCCA using the immune-checkpoint DNA vaccine from the rat iCCA model. This model is also used to evaluate the novel immune therapy for iCCA.

## 2. Materials and Methods

### 2.1. Vector Construction

The PD-1 DNA vaccine was generated by inserting the human IL2 DNA sequence (ATGTATAGGATGCAACTGCTGTCTTGCATTGCTCTGTCTCTGGCACTGGTCACTAACTCTGCC) and mouse *pdcd1* nt 364-1494 into the pVAX vector. The mCTLA4-PD-L1 DNA vaccine was generated by inserting the human IL2 protein sequence (MRRMQLLLLIALSLALVTNS) for enhancing protein secretion [28], mouse *ctla4* nt 316-14449, and mouse *cd274* nt 163-1143 into the pVAC1 vector (Appendix A); mGM-CSF-mEGF is a fusion protein. EGF activates EGFR-mediated signals to promote tumor progression in cholangiocarcinoma [29]. The EGF antibody was produced in rats receiving EGF proteins to influence EGF-mediated EGFR signals, preventing tumor progression. In our experiments, the mGM-CSF-mEGF protein acted as a positive control for the suppression of tumorigenesis in rats. The similarities of PD-1, PD-L1, and CTLA4 nucleotide sequences between mice and rats were analyzed (Appendix A). The purity of pVAX1-hIL2ss-mPD-1 or pVAC-hIL2ss-mCTLA4-mPD-L1 was determined by the *OD*260/*OD*280 ratio, and agarose gel electrophoresis and the accuracy of DNA sequences were determined by DNA sequencing. The expression and secretion of proteins from CHO cells transiently expressing CTLA4-PD-L1 or PD-1 were detected by ELISA (Appendix A).

### 2.2. Thioacetamide (TAA)-Induced iCCA Rat Model

A TAA-induced iCCA rat model that recapitulated the multistage progression of human iCCA was used [22]. Briefly speaking, male Sprague-Dawley rats fed with drinking water with TAA 300 mg/L were used in this study. After 30 weeks of feeding with water containing TAA, animal positron emission tomography (PET) was performed to measure the baseline relative standardized uptake value (SUVr) of the tumors in the rats. Animal experiments were approved by the Institutional Animal Care and Utilization Committee of Chang Gung Memorial Hospital, Linkou (IACUC no. 2018092502 and 2015121001).

### 2.3. Immunization of Rat with DNA Vaccines

A mixture of 300 μg of DNA diluted with 5% dextrose in water (*w*/*v*) to a final volume of 300 μL was gently mixed with 300 μL of liposome and incubated for 25 min at room temperature. The mixture was injected into the muscle at multiple sites (four limbs). Dioleoyl-3-trimethylammonium propane (DOTAP) cholesterol liposomes used in this study were prepared according to the protocol described by Templeton N.S. et al. [30] for liposome QC. In brief, we examined its activity and that of commercially acquired lipofectamine by transfecting CHO cells into a 6-well plate with 0.5 μg of pCNDA-CMV-luciferase. Two days after transfection, cells were subjected to a luciferase assay using a substrate from Promega. Luciferase activities from transfected cells using lipofectamine and DOTAP: cholesterol liposomes were comparable.

### 2.4. Detection of Serum Antibodies against CTLA-4

The 0.5 μg/mL of mCTLA4 (mPD-1, or mPD-L1) in phosphate buffered saline (PBS) was coated to a 96-well 442,404 NUNC-IMMUNO plate (Thermo Fisher Scientific Inc., Waltham, MA, USA) 50 uL/well at 4 °C overnight. After 1 h of blocking (1% BSA in PBS), rat serum 1:25 diluted with reagent diluent (0.1% BSA, 0.05% Tween-20 in 20 mM Tris-base, 150 mM NaCl, pH 7.2–7.4) was added to a plate 100 μL/well and incubated for 2 h at room temperature. Goat antirat IgG HRP with reagent diluent (1:5000) 100 μL/well was applied as a secondary antibody for 1 h at room temperature. TMB (Tetramethylbenzidine) substrate (50 μL/well) was applied for 20 min and then stopped with 1 M H2SO4 (50 μL/well). Absorbance at 450 nm was read with a TECAN infinite M200PRO plate reader. Moreover, the plate was washed three times with 0.05% Tween-20 in PBS between each step.

### 2.5. Immunohistochemistry (IHC), and Hematoxylin and Eosin (H&E) Staining

CD8 and PD-L1 expression was also examined in thioacetamide (TAA)-induced iCCA in rat tissues. Hepatectomy specimens from all rats were fixed in formalin and embedded in a 4 μm paraffin section and then stained for selected markers. Primary antibodies were incubated overnight at 4 °C (anti-PD-L1, bs-10159R 1:800, Bioss Inc., Woburn, MA, USA; and anti-CD8, GTX41831 1:200, GeneTex Inc., Irvine, CA, USA). Control slides were simultaneously incubated in diluent without the primary antibody. Slides were then washed three times for 5 min each in TBST (Tris Buffered Saline with Tween 20) prior to visualization using the REAL EnVision Detection System, Peroxidase/DAB+, Rb/Mo (K500711, DAKO, Agilent Technologies, Inc., Santa Clara, CA, USA). After 3 TBST washes for 5 min each, slides were counterstained with hematoxylin, mounted, and then blindly analyzed under microscopy. Adjacent H&E staining was also applied to confirm the histological structure. PD-L1 intensity was defined as follows: 0, no staining; 1, very weak staining; 2, weak staining; and 3, strong staining by image J from 0.09 mm^2^ random fields. The numbers of CD8^+^ cells were determined from 0.09 mm^2^ random fields. H score was calculated by multiplying intensity level with the percentage of the positive area.

### 2.6. Evaluation of Treatment Efficacy in Rats Using Positron Emission Tomography (PET)

To evaluate changes in glycolysis in live animals with liver tumors, we conducted 2-deoxy-2-[F-18] fluoro-d-glucose (FDG) PET studies in rats at the Molecular Imaging Center of Chang Gung Memorial Hospital. Overall, a total of 20 rats were treated with TAA and subjected to serial PET scanning in Weeks 21, 23, and 25 using the Inveon™ system (Siemens Medical Solutions USA Inc., Knoxville, TN, USA). Equal numbers of animals were assigned to the control and treatment groups on the basis of their baseline PET results. This ensured that the control and treatment groups possessed similar PET-positive rates. The details of radioligand preparation, scanning protocols, and determination of optimal scanning time were previously described by our group [12,13]. Briefly, the animals were fasted overnight prior to scanning. At 90 min post-^18^F-FDG injection (i.v.), 30 min static scans were performed on all animals. All imaging studies were performed by using a temperature (set to 37 °C) and anesthesia (2% isoflurane vaporized in 100% oxygen)-controlled imaging bed (Minerve System, Esternay, France). PET images were reconstructed using the 2D ordered subset expectation–maximization method (4 iterations and 16 subsets) without attenuation and scatter corrections. All imaging data were processed using the PMOD image analysis workstation (PMOD Technologies Ltd., Zurich, Switzerland). The largest liver tumor for each animal was identified by careful investigation of all three image sets for each rat. The uptake was 18 F-FDG by the largest liver tumor, and the apparent normal liver tissue was quantified by calculating the standardized uptake value (SUV). These values were calculated according to the recommendations of the European Organization for Research and Treatment of Cancer. Tumor regions of interest (ROIs) were determined by using the transverse images of the selected tumors and measuring the largest diameter. Normal liver ROIs were determined by also using the same transverse images. Moreover, mean SUV (SUV_mean_) of the normal liver and tumor tissue was determined, and the tumor-to-liver radioactivity ratio was calculated for comparison. Values of the relative SUV (SUVr) in each rat were determined with SUV_mean_ values on the 5th and 9th weeks, and were normalized with the SUV_mean_ baseline.

### 2.7. PET to Evaluate iCCA Tumor Growth

Details of the PET scans were described in our previous study [31]. Animal PET images were collected to assess the baseline tumors, and every 4 weeks after initiating treatment on the basis of highest tumor-to-liver (T/L) ratio using time–activity curves.

### 2.8. Data Statistics

Data are presented as mean ± SD or mean ± SEM. Differences between experiment and control animals were calculated using the Mann–Whitney U or Student’s *t* tests. Differences in SUV values between experiment and control animals were calculated using one-way ANOVA. A value of *p* ≤ 0.05 was considered to be statistically significant.

## 3. Results

### 3.1. Immune-Cell Infiltration in Rat iCCA

According to previous reports in patients, a subset of CCAs displays high immune-cell infiltration [32]. In this study, we used TAA-administered male Sprague-Dawley (SD) rats to study the effect of the DNA of immune-checkpoint proteins on TAA-induced CCA. SD rats were administered with 300 mg/L TAA for 30 weeks, and invasive CCAs were detected in 100% of TAA-administered SD rats [22]. Thus, clarification of the immune-cell compositions of the tumor microenvironment was the priority. SD rats were administered with water containing 300 mg/L TAA daily, and developed orthotopic rat CCA, illustrating several immune-cell infiltrations, including CD8^+^ T cells (Figure 1A,C). During iCCA tumorigenesis of the rat model, the intensity of PD-L1 expression was also increased on the 27th week (Figure 1B), indicating that PD-L1 expression is critical in iCCA tumorigenesis, and that the experimental model is suitable for studying the immune response in iCCA using the immune-checkpoint DNA vaccine. Furthermore, the infiltration of CD8^+^ immune cells increased as rat CCA progressed on the 27th week (Figure 1C).

### 3.2. mPD-1 DNA Vaccine

TAA-induced iCCA rats were scanned using PET before the starting treatment in order to measure and record baseline tumor volumes. Moreover, rats were divided into two groups according to tumor volumes measured by PET to ensure that the rats in both groups had similar tumor volumes, followed by the collection of the first serum for the baseline antibodies. The mPD-1 DNA vaccine and control serum were injected into rats every week for three consecutive weeks. The second PET was performed on Week 5 to evaluate the tumor response to the DNA vaccine, and the second and third serums, which were collected on Weeks 6 and 10 (Figure 2A). The serum titer of the anti-mPD-1 antibody was measured, and the antibody titer ratio was calculated after normalization to the first serum collection (Figure 2B,C). Furthermore, no increase in antibody levels was noted after mPD-1 DNA vaccination, indicating that the mPD-1 DNA fragment failed to demonstrate its immunogenicity in the TAA-induced iCCA model. However, tumor intensity did not significantly decrease after the mPD-1 DNA fragment treatment; this might have been because of the nonimmunogenicity of mPD-1 DNA fragment (Figure 2D).

### 3.3. Immunogenicity of mCTLA4-PD-L1 DNA Vaccine

Granulocyte–macrophage colony-stimulating factor (GM-CSF) is a glycoprotein cytokine secreted by mononuclear leukocytes. GM-CSF induces several effects on the immune system, including dendritic-cell maturation, macrophage activation, neutrophil proliferation, and T-cell activation. GM-CSF stimulates antigen-presenting cells and disproportionally enhances overall immunity against various endogenous antigens [33,34,35]. In an experiment (Figure 3A), the mCTLA4–PD-L1 DNA vaccine, mGM-CSF-mEGF protein, and negative control solvent were injected into rats on Weeks 1–3. The second and third PET scans were then performed to evaluate the tumor response on the DNA vaccine on Weeks 5 and 9, whereas the second and third serum collections were performed at Weeks 6 and 10 (Figure 3A). The serum anti-mPD-L1 antibody was measured, and the antibody titer ratio was calculated after normalization to the first serum collection (Figure 3B,C). Increased anti-mPD-L1 antibody levels were noted after mCTLA4–PD-L1 DNA vaccination, indicating that the mPD-L1 DNA fragment showed its immunogenicity in the TAA-induced iCCA model. Similarly, anti-mCTLA4 antibody levels increased after mCTLA4-PD-L1 DNA vaccination (Figure 3D,E).

### 3.4. Tumor Response of mCTLA4-PD-L1 DNA Vaccine

Tumor intensity increased in the control group, but decreased after mCTLA4–PD-L1 DNA vaccination treatment on Weeks 5 and 9 (Figure 4A). The infiltration of CD8^+^ T cells increased after mCTLA4–PD-L1 DNA vaccination (Figure 4B). In contrast, the expression of PD-L1 was suppressed after DNA vaccination (Figure 4C). Interestingly, tumor intensity further decreased on Week 9 as compared to that on Week 5 among rats undergoing mCTLA4–PD-L1 DNA vaccination, but not those on the mGM-CSF-mEGF protein treatment. This could be attributed to the dual effect of anti-PD-L1 and anti-CTLA4 antibodies in rats undergoing mCTLA4–PD-L1 DNA vaccination.

## 4. Discussion

The aim of this study was to evaluate the feasibility and efficacy of the immune checkpoints of DNA cancer vaccines in the iCCA rat model. Moreover, T-lymphocyte infiltration was observed in the microenvironment around iCCA cancer cells, and PD-L1 expression was associated with iCCA tumorigenesis of the iCCA rat model, indicating the critical role of immunosuppression in the TAA-induced iCCA rat model. Experiments also found that DNA vaccination targeted CTLA-4 and PD-L1, but not PD-1, eliciting serum-specific antibody production and suppressing the tumor growth in iCCA rats.

To validate these sequences in the de novo animal cancer model, which is close to the human tumor, a TAA-induced iCCA rat model was used. As more than 90% similarity was found between rat and mouse, we directly used these available mouse sequences in the rat model; promisingly, these DNA vaccines induced antibodies resulting in tumor inhibition. Therefore, DNA vaccines may work for in vivo studies. In the future, we will directly compare the mouse and rat sequences to see if rat sequences have superior activity compared to that of mouse sequences.

Although there were higher antibody titers against CTLA4 at the 5th and 9th week time point in mGM-CSF-mEGF fusion-protein-vaccinated mice, there was no detectable increase in the titers of PD-L1 antibody compared with those treated by PBS. GM-CSF is considered an immunological adjuvant, and its fusion to antigens has been adopted in multiple studies [36]. It is inconclusive whether the mGM-CSF-mEGF fusion protein, which is meant to be a negative control of the CTLA4–PD-L1 DNA vaccine, could specifically induce anti-CTLA4 antibodies or GM-CSF moiety to enhance the immune system of the experiment’s mice, thereby distinctively triggering immune response to different proteins, such as CTLA4 and PD-L1 in this study.

The ultimate goal of DNA vaccines is to clinically develop innovative anticancer therapeutics. Sequence similarities between human and mouse or rat PD-L1, CTLA4, and PD-1 are about 75%–80%. Thus, hPD-L1–CTLA4 DNA may be suitable for a clinical trial. This preclinical study provided crucial information. As we established a good in vivo model to examine the efficacy of immune-checkpoint blockade in iCCA, further research should focus on combination therapy consisting of DNA vaccines and other cytotoxic agents or targeted therapy. Therefore, future research would be applied in clinical trials.

## 5. Conclusions

In conclusion, we investigated the immunogenicity and efficacy of immune-checkpoint-based DNA cancer vaccines, and showed that the mCTLA4–PD-L1 DNA vaccine significantly elicited antibody production and demonstrated its antitumor activity.

## Figures and Tables

**Figure 1 vaccines-08-00703-f001:**
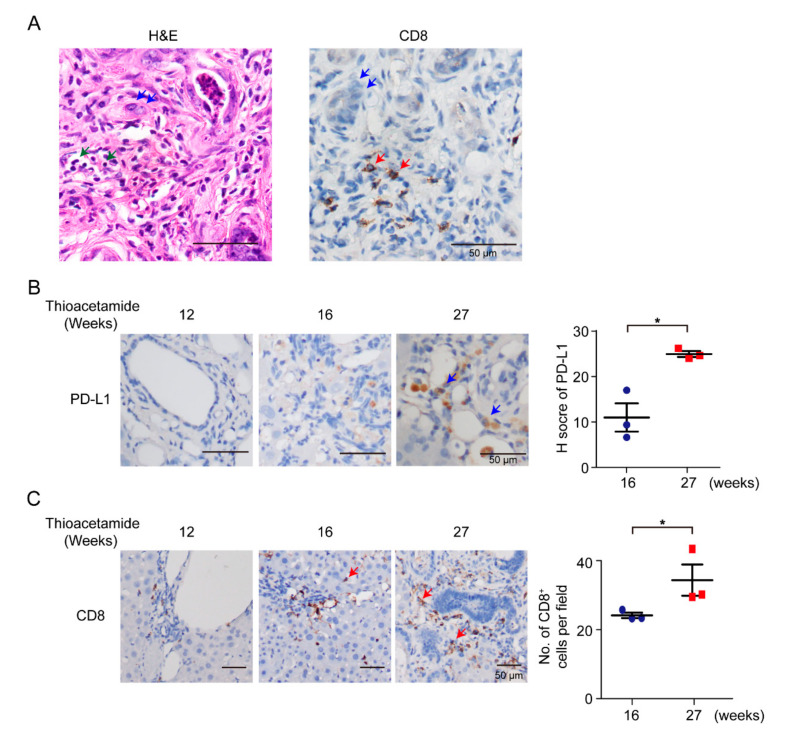
Infiltration of immune cells in rat cholangiocarcinoma (CCA). (**A**) Hematoxylin–eosin (H&E) and immunohistochemical staining of CD8 in TAA (thioacetamide)-induced intrahepatic CCA (iCCA). Blue arrows, CCAs; green arrows, immune cells; red arrows, CD8^+^ T cells. Scale bars: 50 μm. (**B**) Left: Immunohistochemical staining of PD-L1 in TAA-induced iCCA for 12, 16, and 27 weeks. Blue arrows, CCAs. Scale bars: 50 μm. Right: Distribution of H score for PD-L1 from TAA-induced iCCA rats at 16th and 27th weeks (*n* = 3 rats). Blue circles: 16th week; Red squares: 27th week. Values presented as mean ± SEM. * *p* < 0.05 by Student’s *t* test. (**C**) Left: Immunohistochemical staining of CD8 in TAA-induced iCCA for 12, 16, and 27 weeks. Red arrows, CD8^+^ T cells. Scale bars: 50 μm. Right: Numbers of CD8^+^ cells per field from TAA-induced iCCA rats on 16th and 27th weeks (*n* = 3 rats). Blue circles: 16th week; Red squares: 27th week. Values presented as mean ± SEM. * *p* < 0.05 by Student’s *t* test.

**Figure 2 vaccines-08-00703-f002:**
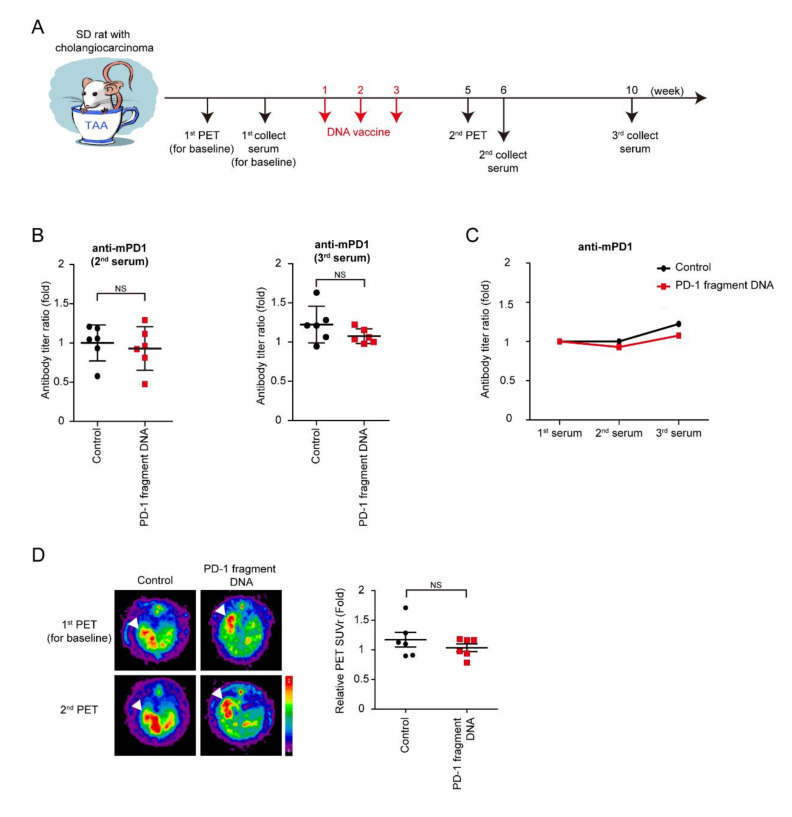
Effect of PD-1 DNA vaccine. (**A**) Schema for experiment design. (**B**) Antibody titers of mouse PD-1 (mPD-1) after 6 (2nd) and 9 (3rd) weeks of PD-1 DNA (Red squares) or control (5% dextrose in water; black circles) compared with those before treatment (1st); *N* = 6. Values presented as mean ± SD; NS: not significant. (**C**) Average values of antibody titers after 6 (2nd) and 9 (3rd) weeks of PD-1 DNA (Red squares) or control (5% dextrose in water; black circles) compared with those before treatment (1st); *N* = 6. (**D**) Left: Representative image of 18F-FDG micro-positron emission tomography (PET) on TAA-induced CCA in Sprague-Dawley (SD) rats. The images were performed before treatment (1st) and after 5 weeks (2nd). Right: The relative of tumor-to-liver (T/L) ratio of standardized uptake value (SUV) in control (5% dextrose in water; black circles) and experiment groups (Red squares) at 5 weeks (2nd) after the experiment. Values presented as mean  ±  SEM. White arrowheads: tumors. *N* = 6. NS: not significant.

**Figure 3 vaccines-08-00703-f003:**
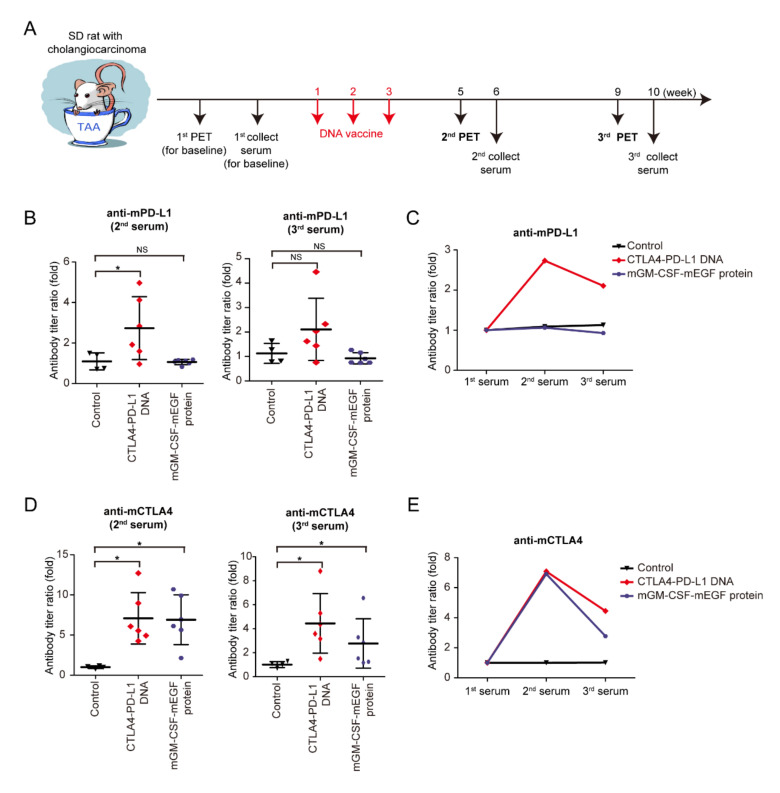
Effect of CTLA4–PD-L1 DNA vaccine. (**A**) Schema for experiment design. (**B**) Antibody titers of mouse PD-L1 (mPD-L1) after 6 (2nd) and 9 (3rd) weeks of CTLA4–PD-L1 DNA (red romboids), mGM-CSF-mEGF protein (blue circles), or control (5% dextrose in water; black triangles) compared with those before treatment; *N* ≥ 4. Values presented as mean ± SD; * *p* < 0.05 by Mann–Whitney U-test; NS: not significant. (**C**) Average values of mPD-L1 antibody titers after 6 (2nd) and 9 (3rd) weeks of CTLA4–PD-L1 DNA (red romboids), mGM-CSF-mEGF protein (blue circles), or control (5% dextrose in water; black triangles) compared with those before treatment; *N* ≥ 4. (**D**) Antibody titers of mouse CTLA4 (mCTLA4) after 6 (2nd) and 9 (3rd) weeks of CTLA4–PD-L1 DNA (red romboids), mGM-CSF-mEGF protein (blue circles), or control (5% dextrose in water; black triangles) compared with those before treatment; *N* ≥ 4. Values presented as mean ± SD; * *p* < 0.05 by Mann–Whitney U-test. (**E**) Average values of mCTLA4 antibody titers after 6 (2nd) and 9 (3rd) weeks of CTLA4–PD-L1 DNA (red romboids), mGM-CSF-mEGF protein (blue circles), or control (5% dextrose in water; black triangles) compared with those before treatment; *N* ≥ 4.

**Figure 4 vaccines-08-00703-f004:**
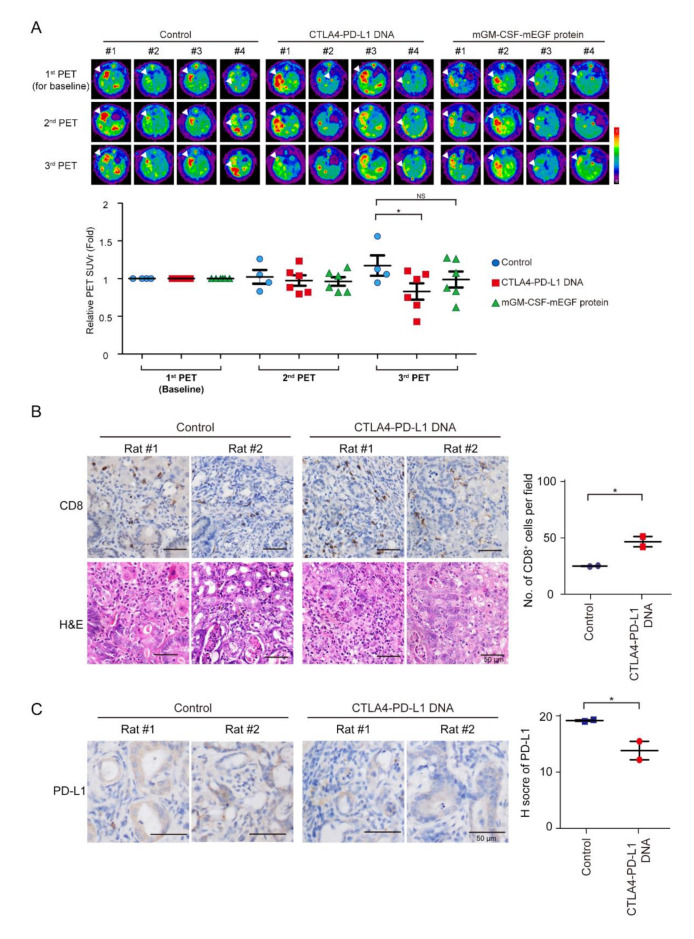
Treatment of CTLA4–PD-L1 DNA impaired tumorigenesis. (**A**) Representative image of 18F-FDG microPET on TAA-induced CCA in SD rats. Images taken before treatment (1st), and after 5 (2nd) and 9 (3rd) weeks of treatment. Change of tumor-to-liver (T/L) ratio of SUV in control (5% dextrose in water; blue circles) and experiment groups at 5 (2nd) and 9 (3rd) weeks after experiment. Red squares: CTLA4-PDL1 DNA; Green triangle: mGM-CSF-mEGF protein. Values presented as mean ± SEM. White arrowheads: tumors; *N* ≥ 4. For third PET, *p* value = 0.031 by one-way ANOVA test and *p* = 0.037 by post hoc comparison using Scheffe test (control versus CTLA4-PD-L1 DNA (*p* = 0.037); control versus mGM-CSF-mEGF protein (*p* = NS); CTLA4–PD-L1 DNA versus mGM-CSF-mEGF protein (*p* = NS); * *p* < 0.05; NS: not significant. (**B**) Left: Hematoxylin–eosin (H&E) and immunohistochemical staining of CD8 in TAA-induced iCCA receiving DNA vaccination or control adjuvant. Scale bars: 50 μm. Right: Numbers of CD8^+^ cells per field from TAA-induced iCCA rats receiving CTLA4–PD-L1 DNA (Red squares) or control solvent (Blue circles); *n* = 2. Values presented as mean ± SEM; * *p* < 0.05 by Student’s *t* test. (**C**) Left: Immunohistochemical staining of PD-L1 in TAA-induced iCCA receiving DNA vaccination or control adjuvant. Scale bars: 50 μm. Distribution of H score for PD-L1 from TAA-induced iCCA rats receiving CTLA4–PD-L1 DNA (Red circle) or control solvent (Blue squares); *n* = 2. Values presented as mean ± SEM; * *p* < 0.05 by Student’s *t* test.

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
