# Peer review of "Comprehensive Evaluation of Immune-Checkpoint DNA Cancer Vaccines in a Rat Cholangiocarcinoma Model"

_vaccines, 2020, doi:10.3390/vaccines8040703_

Round 1

Reviewer 1 Report

In this paper, the authors utilize a TAA-induced iCCA cancer model system in rats. They first detect PD-L1 expression and CD8 T-cell infiltration into the tumor microenvironment by immunohistochemistry (IHC). They showed that a DNA vaccine targeting PD-1 lacked immunogenicity and efficacy. They then showed that a DNA vaccine targeting both PD-L1 and CTLA4 was able to elicit some amount of antibody response,  the vaccine was able to reduce the tumor burden, and by IHC the authors claim that CD8 T-cell abundance went up and PD-L1 expression decreased with vaccine.

This is an interesting topic in a model with important human health implications. The idea of using a vaccine to elicit antibodies de novo against immune checkpoint inhibitors (ICI's) instead of giving the antibody as a therapy is very interesting and novel. Many of the points from the first review were addressed, but others still remain, as outlined below:

Major points:

In figures 1 and 4, the quantification was represented as a combination of technical and biological replicates. The technical replicate, in this setting, is represented by making multiple observations in the same animal, while a biological replicate accumulates results in different subjects, in this case, different rats within the same treatment group.  Averaging technical replicates across different animals is not a sound way to represent the data. Technical replicates should be averaged into a single representative data point for the animal. Then the biological replicates that consist of an average of technical replicates can be compared by statistical tests. If this doesn’t result in significant differences, that is due to the low power of the study with only 2-3 animals being analyzed, and a repeat experiment is required. If significance can be achieved this way with the current data, then a repeat is not required.

In figure 3, the data in part D must be addressed. The discussion is advocating for a specific response to the vaccine. 3D does not show specificity. Why are anti-CTLA4 responses from mice not vaccinated with CTLA4 being elicited? This indicates that there is likely an error in the experimental assay due to some nonspecific interaction, which calls into question the validity of the results.

In figures 3 and 4, data from the GMCSF EGF vaccine is presented as a positive control, but the group is not included in the statistical analysis. These data should be analyzed by ANOVA or similar test to include the variance of the entire dataset. All groups involved in an experiment must be included in the statistical analysis.

In the discussion, the language needs to be toned down to accurately reflect the limitations of the study. For instance, the immunogenicity was not “comprehensively” investigated, especially since T-cell responses were not analyzed. In all cases the number of animals included in experiments is quite small and there is no indication that experiments have been repeated to confirm results.  Therefore, the authors should also clarify that these results are preliminary, and the model needs to be studied further before clinical trials can be attempted.

Minor Points:

In figure S5, all presented lanes need labeling and the ladder needs to be labeled with relevant band sizes.

In line 226, it was mentioned that tumor intensity decreased in the EGF vaccinated group when it did not. It remained stable but did not decrease.

There are some English errors and typos scattered throughout. Please revise for grammar.

In conclusion, this is an interesting study using a relatively novel vaccine concept in an underutilized model that is important for human health.  The manuscript has been improved with the revision. However, the points above must be addressed before the manuscript can be considered scientifically valid and acceptable for publication.

Author Response

Reviewer 1

In this paper, the authors utilize a TAA-induced iCCA cancer model system in rats. They first detect PD-L1 expression and CD8 T-cell infiltration into the tumor microenvironment by immunohistochemistry (IHC). They showed that a DNA vaccine targeting PD-1 lacked immunogenicity and efficacy. They then showed that a DNA vaccine targeting both PD-L1 and CTLA4 was able to elicit some amount of antibody response, the vaccine was able to reduce the tumor burden, and by IHC the authors claim that CD8 T-cell abundance went up and PD-L1 expression decreased with vaccine.

This is an interesting topic in a model with important human health implications. The idea of using a vaccine to elicit antibodies de novo against immune checkpoint inhibitors (ICI's) instead of giving the antibody as a therapy is very interesting and novel. Many of the points from the first review were addressed, but others still remain, as outlined below:

Major points:

Comment 1: In figures 1 and 4, the quantification was represented as a combination of technical and biological replicates. The technical replicate, in this setting, is represented by making multiple observations in the same animal, while a biological replicate accumulates results in different subjects, in this case, different rats within the same treatment group.  Averaging technical replicates across different animals is not a sound way to represent the data. Technical replicates should be averaged into a single representative data point for the animal. Then the biological replicates that consist of an average of technical replicates can be compared by statistical tests. If this doesn’t result in significant differences, that is due to the low power of the study with only 2-3 animals being analyzed, and a repeat experiment is required. If significance can be achieved this way with the current data, then a repeat is not required.

Authors’ reply: Thanks for your comments. We modified the quantification of PD-L1 expression and CD8+ cell numbers according to the suggestion (Response Fig. 1). The data are shown in Figure 1 and Figure 4 in the revised manuscript.

Comment 2: In figure 3, the data in part D must be addressed. The discussion is advocating for a specific response to the vaccine. 3D does not show specificity. Why are anti-CTLA4 responses from mice not vaccinated with CTLA4 being elicited? This indicates that there is likely an error in the experimental assay due to some nonspecific interaction, which calls into question the validity of the results.

Authors’ reply: It could be that GM-CSF stimulates the antigen-presenting cells and enhance overall immunity disproportionally against various endogenous antigens, such as CTLA-4 in our data. The enhanced titer of antibody against CTLA-4 by GM-CSF-EGF was less prominent as CTLA-4-PD-L1 vaccine, suggesting the more specific effect of CTLA-4-PD-L1 DNA vaccine than mGM-CSF-mEGF protein on humoral immunity against CTLA-4. In our future study, it would be more informative if different antigens both related and unrelated to immune checkpoints could be evaluated at a longer time point after immunization.

Comment 3: In figures 3 and 4, data from the GMCSF EGF vaccine is presented as a positive control, but the group is not included in the statistical analysis. These data should be analyzed by ANOVA or similar test to include the variance of the entire dataset. All groups involved in an experiment must be included in the statistical analysis.

Authors’ reply: According to the suggestion, we added the statistical analysis in Figure 3B, 3D, and 4A in the revised manuscript.

Comment 4: In the discussion, the language needs to be toned down to accurately reflect the limitations of the study. For instance, the immunogenicity was not “comprehensively” investigated, especially since T-cell responses were not analyzed. In all cases the number of animals included in experiments is quite small and there is no indication that experiments have been repeated to confirm results. Therefore, the authors should also clarify that these results are preliminary, and the model needs to be studied further before clinical trials can be attempted.

Authors’ reply: Thanks for your critical comments. We removed the “comprehensively” word. In this study, we firstly showed that CTLA-4-PD-L1 DNA can suppress iCCA tumor growth in the rat model and we will further investigate whether CTLA-4-PD-L1 DNA can prevent tumorigenesis in iCCA in rats. Therefore, we will further confirm the results before clinical trials.

Minor Points:

Comment 1: In figure S5, all presented lanes need labeling and the ladder needs to be labeled with relevant band sizes.

Authors’ reply: According to the suggestion, the band sizes and the labels of all lanes were added in Figure S5A in the revised manuscript.

Comment 2: In line 226, it was mentioned that tumor intensity decreased in the EGF vaccinated group when it did not. It remained stable but did not decrease.

Authors’ reply: Thanks for your comments. We removed the description of GM-CSF-mEGF protein and the sentence was modified in the revised manuscript.

Comment 3: There are some English errors and typos scattered throughout. Please revise for grammar.

Authors’ reply: Thanks for your comments. We carefully checked and modified the English errors and typos in the revised manuscript.

Comment 4: In conclusion, this is an interesting study using a relatively novel vaccine concept in an underutilized model that is important for human health. The manuscript has been improved with the revision. However, the points above must be addressed before the manuscript can be considered scientifically valid and acceptable for publication.

Authors’ Reply: We thank you for the positive comment of the Reviewer.

Reviewer 2 Report

In their revised manuscript entitled «Comprehensive evaluation of immune checkpoints DNA cancer vaccines in a rat cholangiocarcinoma model», Pan and colleagues efficiently address all my previous comments, therefore I recommend acceptance of the manuscript at this point.

Author Response

Reviewer 2

In their revised manuscript entitled «Comprehensive evaluation of immune checkpoints DNA cancer vaccines in a rat cholangiocarcinoma model», Pan and colleagues efficiently address all my previous comments, therefore I recommend acceptance of the manuscript at this point.

Authors’ Reply: We thank you for the positive comment of the Reviewer.

Reviewer 3 Report

In this study, the authors use an interesting concept — DNA vaccination against immunological checkpoint molecules PD-1, PD-L1, and CTLA-4 — on a chemical-induced iCCA model in rats.

The methods section has several issues including the omission of important details. For example: is the IL-2 DNA fragment a signal sequence, or protein coding?; is mGM-CSF-mEGF a fusion protein?–and how is it obtained, or produced and how administered?; Explain better, and not in the methods section, why/how this is this a suitable positive control, please. Relatedly, what is a description like the one in lines 70-75 doing in a materials/methods section?; something almost trivial but still: how exactly were mouse/rat nucleotide sequences compared?; lines 92-94: at what stage of cancer induction (weeks after TAA start)?; what adjuvants are used? What liposome was used?; unclear what SUVr is and when/how the tumor/liver ratio is used for comparison.

About the validity of the model: no IHC biomarkers are presented. Why not a] use CK19, as in doi:10.1093/carcin/bgh037, or MUC1/MMP2/MMP9, as in doi:10.1097/01.sla.0000129492.95311.f2—and b] combine such biomarkers with PD-L1 and CTLA-4 staining?

Moreover, I am confused about the PET readout measure and (perhaps mistakenly) conclude that the study does not show tumour progression in the control group, as relative PET activity (SUV) barely goes up 1.5 fold in 10 weeks. That is, if the interpretation from fig 4A is correct: it does not say what the measurements are relative to. Is it the baseline of the individual mouse?  If so, the control group shows no increase in the first 5 weeks, and—except for one mouse—also no increase in the remaining 4 weeks. If not, authors should show the baseline values, both absolute and 'normalized'. If, however, the measure is not a normalization but instead the tumor-to-liver ratio (is that SUVr?) then perhaps my interpretation is too harsh, even though I cannot judge such a ratio without baseline comparison.

Whatever the case, I need to see tumor progression, relative to start of treatment, to agree on model validity. I would also stress the need to report progression-free survival if that is feasible, or do an assessment of tumour size, number or pathology after treatment. Quantifications of T cells may mean more if a time point was mentioned.

As presented, this model does not convincingly reflect the aggressive cancer growth in patients, rendering it (or the main readout measure) unsuitable to asses efficacy of immunotherapy for this indication. In that light, the claim made (line 229-233) that the dual vaccination attempt led to tumour intensity decrease (which perhaps it did for up to 3 mice) is not sufficient to show efficacy. Also the claim that this supposed effect is due to both antibodies cannot be made without, e.g., single vaccine controls.

Points of moderate concern:

  • If we would still take the antibody titer results at face value, and expect at least a 4-fold increase in Ab titer following a successful vaccination (only seen in Fig 3D, E: how do the authors explain the increase of anti-CTLA-4 titer following the injection (?) of an unrelated (fusion?) protein?
  • Line 163: how does positive/increased staining for PD-L1 indicate any (critical) role in tumorigenesis?
  • The number of rats used for image quantifications (e.g. Fig 4) is low, as statistics should be done on the level of the animal, not fields-of-view.
  • The writing also needs improvement, with some English language issues (use of 'the', for instance) and the introduction and discussion are not written well. Lines 256-278 should be moved up into the introduction, and the rest of the discussion is not very strong: it contains mention of unspecified, irrelevant preliminary data (lines 279-283), illogical statements suggesting that DNA sequence homology promises successful therapeutic translation from rodent to patients, and the unfounded statement that this work comprehensively investigated the immunogenicity and efficacy of this approach. 
  • What is H score? What is orthotropic cancer?

Author Response

Reviewer 3

In this study, the authors use an interesting concept — DNA vaccination against immunological checkpoint molecules PD-1, PD-L1, and CTLA-4 — on a chemical-induced iCCA model in rats.

Comment 1: The methods section has several issues including the omission of important details. For example: is the IL-2 DNA fragment a signal sequence, or protein coding?; is mGM-CSF-mEGF a fusion protein?–and how is it obtained, or produced and how administered?; Explain better, and not in the methods section, why/how this is this a suitable positive control, please. Relatedly, what is a description like the one in lines 70-75 doing in a materials/methods section?; something almost trivial but still: how exactly were mouse/rat nucleotide sequences compared?; lines 92-94: at what stage of cancer induction (weeks after TAA start)?; what adjuvants are used? What liposome was used?; unclear what SUVr is and when/how the tumor/liver ratio is used for comparison.

Authors’ reply:

1) IL2ss represents the signal sequence of IL2, which is composed of 20 amino acids and was cloned at the N-terminal of the coding sequences of the DNA vaccines used in this manuscript. According to a previous study (J Gene Med. 2005 Mar;7(3):354-65.), hIL2ss was incorporated into chimeric protein for enhancing secretion.

2) mGM-CSF-mEGF is a fusion protein. GM-CSF induces several effects on the immune system, including dendritic cell maturation, macrophage activation, neutrophil proliferation, and T cell activation. GM-CSF stimulates the antigen presenting cells and enhances overall immunity disproportionally against various endogenous antigens (J Immunother Cancer. 2014 May 13;2:11; Blood. 2008 Jan 15;111(2):485-91; Blood. 2009 Aug 13;114(7):1289-98). EGF activates EGFR-mediated signals to promote tumor progression in cholangiocarcinoma (J Hepatol. 2014 Aug;61(2):325-32.). The antibody of EGF was produced in the rats receiving EGF proteins to influence the EGF-mediated EGFR signals, preventing tumor progression. In our experiments, mGM-CSF-mEGF protein acted as a positive control for the suppression of tumorigenesis in rats.

3) According to the suggestions, we moved the paragraph (line 70-75) to the Result section in the revised manuscript.

4) We used the mouse DNA sequences in this study. We wanted to demonstrate that the similarities of PD1, PD-L1, and CTLA4 were high.

5) In line 92-94, SD rats were administered with TAA 300 mg/L for 30 weeks and the invasive CCAs were detected in 100% in TAA-administered SD rats (Carcinogenesis. 2004 Apr;25(4):631-6)

6) The adjuvant is 5% dextrose in water (w/v).

7) DOTAP: cholesterol liposomes used in this study were prepared according to the protocol described by Templeton, N.S. et. al. (Nat Biotechnol 1997;15:647-652). For liposome QC. In brief, we examined its activity and that of commercially acquired lipofectamine by transfecting CHO cells in a 6-well plate with 0. 5 μg of pCNDA-CMV-luciferase.  Two days after transfection, the cells were subjected to luciferase assay using the substrate from Promega. The luciferase activities from transfected cells using lipofectamine and DOTAP: cholesterol liposomes were comparable. 

8) The mean SUVr (SUVmean) of the normal liver and tumor tissue was determined, and the tumor-to-liver radioactivity ratio was calculated for comparison. The values of the relative SUVr in each rat were determined the values of SUVmean at the 5th and 9th weeks and normalized itself SUVmean baseline.

Comment 2: About the validity of the model: no IHC biomarkers are presented. Why not a] use CK19, as in doi:10.1093/carcin/bgh037, or MUC1/MMP2/MMP9, as in doi:10.1097/01.sla.0000129492.95311.f2—and b] combine such biomarkers with PD-L1 and CTLA-4 staining?

Authors’ reply: In our previous publication (Carcinogenesis. 2004 Apr;25(4):631-6), the expressions of CK19, c-Met, and c-Erb-B2 expression in TAA-induced iCCA have been demonstrated. The represented images of CK19 in different fields were shown. (Please see the attachment)

Comment 3: Moreover, I am confused about the PET readout measure and (perhaps mistakenly) conclude that the study does not show tumour progression in the control group, as relative PET activity (SUV) barely goes up 1.5 fold in 10 weeks. That is, if the interpretation from fig 4A is correct: it does not say what the measurements are relative to. Is it the baseline of the individual mouse?  If so, the control group shows no increase in the first 5 weeks, and—except for one mouse—also no increase in the remaining 4 weeks. If not, authors should show the baseline values, both absolute and 'normalized'. If, however, the measure is not a normalization but instead the tumor-to-liver ratio (is that SUVr?) then perhaps my interpretation is too harsh, even though I cannot judge such a ratio without baseline comparison.

Authors’ reply: In the TAA-induced iCCA, the tumor growth was slow and the rate of iCCA tumor growth has been demonstrated in our previous reports (Oncol Lett. 2018 Jul; 16(1): 566–572; Oncotarget. 2014 May; 5(9): 2372–2389; Cancer. 2013 Jan 15;119(2):293-303). According to our report (Mol Imaging Biol. Jul-Aug 2008;10(4):209-16), animal PET is feasible for the detection of CCA in rats. The mean SUVr (SUVmean) of the normal liver and tumor tissue was determined, and the tumor-to-liver radioactivity ratio was calculated for comparison. The values of the relative SUVr in each rat were determined the values of SUVmean at the 5th and 9th weeks and normalized itself SUVmean baseline.

Comment 4: Whatever the case, I need to see tumor progression, relative to start of treatment, to agree on model validity. I would also stress the need to report progression-free survival if that is feasible, or do an assessment of tumour size, number or pathology after treatment. Quantifications of T cells may mean more if a time point was mentioned.

Authors’ reply: Thanks for your critical comments. Indeed, progression-free survival (PFS) can provide more information on tumor progression. However, the animal welfare in our hospital does not allow long-term observation and PFS was not shown in our previous reports (Oncol Lett. 2018 Jul; 16(1): 566–572; Oncotarget. 2014 May; 5(9): 2372–2389; Cancer. 2013 Jan 15;119(2):293-303).

Comment 5: As presented, this model does not convincingly reflect the aggressive cancer growth in patients, rendering it (or the main readout measure) unsuitable to asses efficacy of immunotherapy for this indication. In that light, the claim made (line 229-233) that the dual vaccination attempt led to tumour intensity decrease (which perhaps it did for up to 3 mice) is not sufficient to show efficacy. Also the claim that this supposed effect is due to both antibodies cannot be made without, e.g., single vaccine controls.

Authors’ reply: Although the TAA-induced iCCA was not aggressive cancer, the CTLA4-PD-L1 DNA vaccine treatment suppressed the tumor growth, and the antibodies of CTLA4 and PD-L1 were upregulated. In previous studies (Carcinogenesis. 2004 Apr;25(4):631-6; Ann Surg. 2004 Jul;240(1):89-94), rat TAA-ICCs have been compared to human ICCs. In histologic features, multifocal bile ductular proliferation with intestinal metaplasia and increased histologic atypia were observed in SD rats were administered with TAA 300 mg/L for 9 weeks. At the 16th week, cytokeratin (CK19)-expressing invasive intestinal-type CCA with stromal desmoplasia was obvious. The invasive CCAs were detected in 100% in TAA-administered SD rats at the 22nd week. Thus, this TAA-induced iCCA model can mimic the multi-step model of human CCA so this model is a good candidate for testing DNA vaccination before entering clinical trials. In our preliminary data from the one of the corresponding authors, Keng-Li Lan, the growth of murine CT26 colorectal tumors was reduced in the mice receiving only PD-L1 DNA or CTLA4 DNA. However, the effects of PD-L1 DNA or CTLA4 DNA on the tumor growth suppression were less than that of CTLA4-PD-L1 DNA on the tumor suppression in the mouse model. Thus, we used CTLA4-PD-L1 DNA to investigate the effect of CTLA4-PD-L1on TAA-induced iCCA in the rat model.

Points of moderate concern:

Comment 1: If we would still take the antibody titer results at face value, and expect at least a 4-fold increase in Ab titer following a successful vaccination (only seen in Fig 3D, E: how do the authors explain the increase of anti-CTLA-4 titer following the injection (?) of an unrelated (fusion?) protein?

Authors’ reply: It could be that GM-CSF stimulates the antigen presenting cells and enhance overall immunity disproportionally against various endogenous antigens, such as CTLA-4 in our data. The enhanced titer of antibody against CTLA-4 by GM-CSF-EGF was less prominent as CTLA-4-PD-L1 vaccine, suggesting the more specific effect of CTLA-4-PD-L1 DNA vaccine than mGM-CSF-mEGF protein on humoral immunity against CTLA-4. In our future study, it would be more informative if different antigens both related and unrelated to immune checkpoints could be evaluated at a longer time point after immunization.

Comment 2: Line 163: how does positive/increased staining for PD-L1 indicate any (critical) role in tumorigenesis?

Authors’ reply: In our data, the expression of PD-L1 was upregulated at the 27th week. However, no evidence can provide the critical role of PD-L1 in tumorigenesis. Thus, we removed the description of that in the result section in the revised manuscript.

The modified sentence:

“During iCCA tumorigenesis of the rat model, the intensity of the PD-L1 expression was also increased at the 27th week (Figure 1B), indicating that the experimental model is suitable for studying the immune response in iCCA using the immune checkpoint DNA vaccine.”

Comment 3: The number of rats used for image quantifications (e.g. Fig 4) is low, as statistics should be done on the level of the animal, not fields-of-view.

Authors’ reply: Thanks for your comments. We modified the quantification of PD-L1 expression and CD8+ cell numbers according to the suggestion (Response Fig. 2-Please see the attachment). The data are shown in Figure 1 and Figure 4 in the revised manuscript.

Comment 4: The writing also needs improvement, with some English language issues (use of 'the', for instance) and the introduction and discussion are not written well. Lines 256-278 should be moved up into the introduction, and the rest of the discussion is not very strong: it contains mention of unspecified, irrelevant preliminary data (lines 279-283), illogical statements suggesting that DNA sequence homology promises successful therapeutic translation from rodent to patients, and the unfounded statement that this work comprehensively investigated the immunogenicity and efficacy of this approach. 

Authors’ reply: According to the suggestions, we moved the paragraph (line 256-265) to the introduction section and removed the paragraph (line 266-288).

Comment 5: What is H score? What is orthotropic cancer?

Authors’ reply:

  1. The H score was calculated by multiplying the intensity level and the percentage of the positive area and the description was added in the materials and methods section in the revised manuscript.
  2. The word “orthotropic" is a typo. We used the word “orthotopic” for correction in the revised manuscript

Round 2

Reviewer 1 Report

Please see PDF attached.

Author Response

Thank you for your work in your revision. The manuscript has been improved by your efforts. However, more revision is required.

Major points:

1. This document is very difficult to read with it being a pdf of a word document with track changes. Please re-send as a clean pdf (no track changes) with places you have altered highlighted.

Authors’ reply: We apologize for any inconvenience caused. The clean PDF file was uploaded in this revision.

2. Changing the dot plots to bar graphs in Fig 1 and 4 with identical averages to the previous draft makes me think that the issue was not resolved but that the presentation was changed to mask the issue. To remind you, the issue originally was that all the technical replicates (fields) from the same animal need to be averaged into one final data point. Artificially inflating the sample size by using technical replicates in your analysis is a fallacy called pseudo replication (https://www.ncbi.nlm.nih.gov/pmc/articles/PMC5902037/). There should only be the number of data points on the graph equal to the number of animals analyzed. If that leads to data that is not significant, then it must be reported as such or the experiment needs to be repeated to increase the statistical power. Please change the graph type back to dot plots.

Authors’ reply: Thanks for your suggestion. We modified the bar graphs to the dot plots in Figure 1 and Figure 4 in the revised manuscript.

3. In figure 3, we discussed previously about how anti-ctla4 responses in the GMCSF-EGF vaccine showed non-specificity in the system. The language of the results was toned down and has included more background information on GM-CSF. However, you should expand on your qualifying statements explaining the theoretical nature of the system. You simply cannot conclude from this dataset that vaccine-specific antibodies to CTLA4 are being induced. It needs to be abundantly clear to the reader that the CTLA4 data is inconclusive and needs further study.

Authors’ reply: Although there are higher titers of antibody against CTLA4 at the 2nd week time point in the mGM-CSF-mEGF fusion protein vaccinated mice, there is no detectable increase of the titers of PDL1 antibody as compared with those treated by PBS. It is inconclusive that whether mGM-CSF-mEGF fusion protein, which meant to be a negative control of CTLA4-PDL1 DNA vaccine, could specifically induce anti-CTLA4 antibody or GM-CSF moiety enhance the immune system of the experimental mice, thereby triggering immune response distinctively to different proteins, such as CTLA4 and PDL1 in this study. This description was incorporated into the discussion section (PAGE 16)

Minor points:

1. As mentioned previously, datasets need to be more fully analyzed by statistics. Particularly in Fig 4a the tumor size measurements should be analyzed by ANOVA in order to compare all groups at once, taking into account the full variance of the system

Authors’ reply: Thanks for your suggestion. We modified the statistical analysis by one way ANOVA test in Figure 4A. For third PET, the P value=0.031 by one way ANOVA test and p= 0.037 by post hoc comparison using Scheffe test (control versus CTLA4-PD-L1 DNA (p=0.037); control versus mGM-CSF-mEGF protein (p=NS); CTLA4-PD-L1 DNA versus mGM-CSF-mEGF protein (p=NS). This description was incorporated into the corresponding figure legends (PAGE 25)

2. It is unclear what the formulation of the vaccines are from the methods. What liposome was utilized for the DNA? What amounts, formulation, and adjuvant (if any) were utilized for the GMCSF-EGF protein vaccine, and what was the quality control for it?

Authors’ reply:

  1. DOTAP: cholesterolliposomes used in this study were prepared according to the protocol described by Templeton, N.S. et. al. (Nat Biotechnol 1997;15:647-652). For liposome QC. In brief, we examined its activity and that of commercially acquired lipofectamine by transfecting CHO cells in a 6-well plate with 0. 5 μg of pCNDA-CMV-luciferase.  Two days after transfection, the cells were subjected to luciferase assay using the substrate from Promega. The luciferase activities from transfected cells using lipofectamine and DOTAP: cholesterol liposomes were comparable. This description was incorporated into the method section (PAGE 8)
  2. We did not include any adjuvant in the GMCSF-EGF vaccine given that GMCSF is considered as an immunological adjuvant and its fusion to antigens had been adopted in multiple studies (BMC Immunol. 2011 Dec 30;12:72).

Reviewer 3 Report

My assessment is that very little has changed regarding my main concerns: the validity of the used model , or of the limited readout measures, for this therapy remains insufficiently resolved.

I had requested that – if indeed the TAA rat assessed by PET was not a progressive iCCA model – additional parameters should be included, such as DFS or histopathological analysis of tumours post treatment, or even just size/number of tumours (to confirm PET measures). I can accept that new animal studies are not a preferred option, and that survival is a discredited experimental endpoint, but I did expect additional analyses of the existing experiments.

What is currently presented: increased titer (although mirrored by a control condition), a marginal decrease of the PET readout and an increase of T cells. I find it a bit minimal, but it can be acceptable on the condition that the limitations of the model are clear to the reader.

So, in conclusion, with regard to experimental clarity, not all of my points have been improved. I appreciate a number of improvements, but some of my remarks went unaddressed altogether, and others were only clarified in the rebuttal, not the manuscript. Authors should at least address all moderate/minor points.

Author Response

My assessment is that very little has changed regarding my main concerns: the validity of the used model, or of the limited readout measures, for this therapy remains insufficiently resolved.

I had requested that – if indeed the TAA rat assessed by PET was not a progressive iCCA model – additional parameters should be included, such as DFS or histopathological analysis of tumours post treatment, or even just size/number of tumours (to confirm PET measures). I can accept that new animal studies are not a preferred option, and that survival is a discredited experimental endpoint, but I did expect additional analyses of the existing experiments.

What is currently presented: increased titer (although mirrored by a control condition), a marginal decrease of the PET readout and an increase of T cells. I find it a bit minimal, but it can be acceptable on the condition that the limitations of the model are clear to the reader.

So, in conclusion, with regard to experimental clarity, not all of my points have been improved. I appreciate a number of improvements, but some of my remarks went unaddressed altogether, and others were only clarified in the rebuttal, not the manuscript. Authors should at least address all moderate/minor points.

Authors’ reply:

  1. We deeply appreciate the reviewer’s comments. We are afraid that the reviewer is not familiar with this TAA-induced rat iCCA. This TAA-induced rat iCCA model has been used since 2004 after the first publication in our group (Carcinogenesis. 2004 Apr;25(4):631-6. doi: 10.1093/carcin/bgh037). This model mimics human iCAA so can be used as an animal model for the study of iCCA. Currently, more than one hundred citations for this important article indicating that this model has been repeatedly validated to serve as a powerful pre-clinical platform for therapeutic and chemoprevention strategies for human iCCA.
  2. The next challenge was to evaluate the tumor response. Unlike the xenograft animal model, the tumor size can be visualized and evaluated by inspection. This TAA-induced iCCA infiltrates the cirrhotic liver of the rats and forms the tumors with various numbers and sizes. In addition, not all the rats have the same number and size of tumors at the beginning of treatment. Therefore, it is impossible to measure baseline tumors except sacrifice the rats or PET/CT scan. We tried to use CT scan or ultrasonography but the tumors cannot be assessed by traditional methods. Therefore, we found PET/CT is a good tool to evaluate the tumor response and the paper was published in 2008. (Mol Imaging Biol. Jul-Aug 2008;10(4):209-16. doi: 10.1007/s11307-008-0141-8. Epub 2008 May 20.) Bisdes, FDG animal PET images corresponded with necropsy. This article has been cited by 25 times so PET/CT is validated.
  3. Some comments were incorporated into the revised manuscript.
  • The description of IL2ss (PAGE 7)
  • The description of liposomes (PAGE 8)
  • The measure of SUVr (PAGE 11)

Round 3

Reviewer 1 Report

The authors have addressed my concerns appropriately. Please go through one more time looking for typos and English grammar errors (for example, lines 764-767 are a run-on sentence). Thank you for all of the work. In my eyes it is ready to be published.

Reviewer 3 Report

I have nothing further to add. I think this is now an editorial decision.

This manuscript is a resubmission of an earlier submission. The following is a list of the peer review reports and author responses from that submission.

Round 1

Reviewer 1 Report

In their manuscript entitled «Comprehensive evaluation of immune checkpoints DNA cancer vaccines in a rat cholangiocarcinoma model» Pan and colleagues investigate the use of DNA vaccines to trigger the production of antibodies against the immune checkpoints PD-1, PD-L1 and CTLA4 in a thioacetamide (TAA)-induced intrahepatic cholangiocarcinoma (iCCA) rat model. The ultimate goal of this study is to evaluate the efficiency of these DNA vaccines on the inhibition of immune-suppressive proteins, such as PD-1, PD-L1 and CTLA4, aiming at strengthening the immune responses against iCCA development. Overall, this is a very interesting and novel study since the current literature on this topic is not yet extended. Experimentally, the study is well-designed, the references adequate and the flow of the text quite easy to read. I only have few minor points that the authors should address in order to enhance further the quality of the paper.

Major comments

  • In Figures 1B, 1C and 4B and 4C quantification of the IHC images should be done in order to assess easier the differences of the staining of PD-L1 and CD8 between the different experimental conditions.
  • The authors have not introduced the mGM-CSF-mEGF protein in their material and methods section and neither they mention the purpose of its usage in their experiment.
  • How the authors explain the effect of mGM-CSF-mEGF protein on the anti-mCTLA4 serum levels in the experiment of the Fig. 3D ?

Minor comments

  • Error bars in the graphs of the Fig. 4A are missing. They should be added.
  • Line 86 : plat (typo)
  • Line 138 : the authors should define SD rats
  • Lines 165-167 : I think that the tumor intensity was decreased but not significantly
  • What is the similarity between human and mouse or rat PD-L1, CTLA4 and PD1 isoforms ? This coud be discussed in the Discussion section.
  • Do the authors believe (or have they tried) that a DNA vaccine against only PD-L1 or only CTLA4 could have similar anti-tumor effects on the rat iCCA model, with their mCTLA4-PD-L1 DNA vaccine ? This possibility could also be discussed in the Discussion section.

Author Response

Major comments

Comment 1: In Figures 1B, 1C, 4B, and 4C quantification of the IHC images should be done in order to assess easier the differences of the staining of PD-L1 and CD8 between the different experimental conditions.

Authors’ reply: As suggested by the reviewer, we have added the quantified data of IHC images, and the data are shown in Figures 1B, 1C, 4B, and 4C in the revised manuscript

Comment 2: The authors have not introduced the mGM-CSF-mEGF protein in their material and methods section and neither they mention the purpose of its usage in their experiment.

Authors’ reply: As suggested by the reviewer, we have added the new paragraph listed below in the material and methods section in the revised manuscript.

“GM-CSF (Granulocyte-macrophage colony-stimulating factor), a glycoprotein cytokine, secreted by mononuclear leukocytes. GM-CSF induces several effects on the immune system, including dendritic cell maturation, macrophage activation, neutrophil proliferation, and T cell activation. GM-CSF stimulates the antigen-presenting cells and enhances overall immunity disproportionally against various endogenous antigens (J Immunother Cancer. 2014 May 13;2:11; Blood. 2008 Jan 15;111(2):485-91; Blood. 2009 Aug 13;114(7):1289-98). EGF activates EGFR-mediated signals to promote tumor progression in cholangiocarcinoma (J Hepatol. 2014 Aug;61(2):325-32.). The antibody of EGF was produced in the rats receiving EGF proteins to influence the EGF-mediated EGFR signals, preventing tumor progression. In our experiments, mGM-CSF-mEGF protein acted as a positive control for the suppression of tumorigenesis in rats.”

Comment 3: How the authors explain the effect of mGM-CSF-mEGF protein on the anti-mCTLA4 serum levels in the experiment of Fig. 3D ?

Authors’ reply:

It could be that GM-CSF stimulates the antigen-presenting cells and enhance overall immunity disproportionally against various endogenous antigens, such as CTLA-4 in our data. The enhanced titer of antibody against CTLA-4 by GM-CSF-EGF was less prominent as the CTLA-4-PD-L1 vaccine, suggesting the more specific effect of CTLA-4-PD-L1 DNA vaccine than mGM-CSF-mEGF protein on humoral immunity against CTLA-4. In our future study, it would be more informative if different antigens both related and unrelated to immune checkpoints could be evaluated at a longer time point after immunization.

Minor comments

Comment 1: Error bars in the graphs of the Fig. 4A are missing. They should be added.

Authors’ reply: Thank you for the suggestion. We have added the error bars in the revised manuscript.

Comment 2: Line 86 : plat (typo)

Authors’ reply: Thank you for the suggestion. We have corrected in the revised manuscript.

Comment 3: Line 138 : the authors should define SD rats

Authors’ reply: Thank you for the suggestion. We have modified the description as “TAA-administered male Sprague-Dawley (SD) rats”

Comment 4: Lines 165-167 : I think that the tumor intensity was decreased but not significantly

Authors’ reply: Thank you for the suggestion. We have modified the description as “However, the tumor intensity did not significantly decrease after the mPD1 DNA fragment treatment.”

Comment 5: What is the similarity between human and mouse or rat PD-L1, CTLA4 and PD1 isoforms? This could be discussed in the Discussion section.

Authors’ reply: The similarities between human and mouse or rat PD-L1, CTLA4, and PD1 are shown below:

  PDCD1 (nt 61-498) CTLA4 (nt 106-483) CD274 (nt 55-381)
Homo sapiens vs Mus musculus 75% 77% 76%
Homo sapiens vs Rattus norvegicus 77% 78% 77%
Mus musculus vs Rattus norvegicus 93% 94% 93%

As suggested by the reviewer, we have added this paragraph shown below in the discussion section in the revised manuscript.

“The ultimate goal of the DNA vaccines is to develop an innovative anticancer therapeutics in clinical. The sequence similarities between human and mouse or rat PD-L1, CTLA4, and PD1 are about 75-80%. Thus, hPD-L1-CTLA4 DNA may be suitable for the clinical trials. This preclinical study provided the crucial information of the future clinical trial.”

Comment 6: Do the authors believe (or have they tried) that a DNA vaccine against only PD-L1 or only CTLA4 could have similar anti-tumor effects on the rat iCCA model, with their mCTLA4-PD-L1 DNA vaccine? This possibility could also be discussed in the Discussion section.

Authors’ reply: As suggested by the reviewer, we have added this paragraph shown below in the discussion section in the revised manuscript.

“In several studies demonstrated that only CTLA4 antibody or only PD-L1 antibody can reduce tumorigenesis. In our preliminary data from the corresponding author Dr. Lan, the growth of murine colorectal tumors was reduced in the mice receiving only PD-L1 DNA or only CTLA4 DNA. However, the effects of PD-L1 DNA or CTLA4 DNA on the tumor growth suppression were less than the effect of CTLA4-PD-L1 DNA on the tumor suppression in the mouse model. Thus, we used CTLA4-PD-L1 DNA to investigate the effect of CTLA4-PD-L1on TAA-induced iCCA in the rat model.”

Reviewer 2 Report

In this paper, the authors utilize a TAA-induced iCCA cancer model system in rats. They first detect PD-L1 expression and CD8 T-cell infiltration into the tumor microenvironment by IHC. They showed that a DNA vaccine targeting PD-1 lacked immunogenicity and efficacy. They then showed that a DNA vaccine targeting both PD-L1 and CTLA4 was able to elicit some amount of antibody response,  the vaccine was able to reduce the tumor burden in at least one rat, and by IHC the authors claim that CD8 T-cell abundance went up and PD-L1 expression decreased with vaccine.

This is an interesting topic in a model with important human health implications. The idea of using a vaccine to elicit antibodies de novo against ICI's instead of giving the antibody as a therapy is very interesting and novel. That being said, I have major reservations that are outlined below.

Introduction:

The introduction is insufficient, lacking essential topics. For instance it lacks the following, which I consider to be important:

1) Background about the vaccine platform. Why was a DNA vaccine chosen if the end goal was antibody production? Other vaccine modalities are generally superior at eliciting antibody responses.

2) Background about the model system, especially what is known about its immunology, its response to ICIs, and how well it parallels the human condition. I was unable to find any papers that have attempted ICI therapy in this model, so it is unknown if the vaccine would be efficacious even if it proves to be immunogenic. Similarly, there isn't much data available on the human cancer and how it reacts to ICI. The authors note that only MSI-high or dMMR varieties are recommended to receive ICI treatment. Also reference 20 that shows modest efficacy is in the supplement of the journal and was not accessible by my institution, so I was unable to confirm that data.

3) A better transition to understanding the overall rationale driving this work. It is unclear why the authors are testing out a vaccine against ICIs instead of testing the efficacy of ICIs themselves. I consider this a major logical flaw that needs to be directly addressed. If ICI's don't work in this model, why would vaccines against ICI's work? If ICI's do work in this model, why would a vaccine be a better idea than using the ICI's themselves? This needs explanation.

Methods:

It is unclear if the IL-12 sequence is the entire protein used as an immune stimulant or if it is the secretion signal only. Section 2.1 on the whole needs revision for clarity.

Why are the authors using mouse sequences in a rat model? What is the rationale? Why not use the rat sequences? Is there a benefit for using a non-native sequence? Is that translatable? Similarly, what is the rationale for using the human version of IL-12?

This paper is missing quality control measures of the DNA vaccine construct. Was the stock pure? Was it sequence verified? Did the authors confirm that it does produce the protein of interest after cellular transfection? Is that protein secreted?

How old were the rats at baseline?

It is unclear if these results were repeated or the result of one in vivo experiment each. At least two independent experiments are necessary to validate results.

Statistical methods are mentioned here, but never mentioned that I saw in the figures or text of the results. This is a glaring flaw

Results:

section 3.3/Fig 1: If the authors make claims that cells are increased, then there must be quantifiable data presented. IHC pictures are valuable, but they must be corroborated with quantifiable data to make claims about increases of cellular infiltrates, otherwise the results may be unintentionally biased. To help with the issue of bias, analysis should also be done in blinded fashion.

Figure 2/3:

Figure 3 has an additional group as compared to figure 2; mGM-CSF-mEGF. This group was not sufficiently introduced and it is unclear what purpose this group serves. I'm assuming it is a control for a vaccine that induces inflammation, but it is not discussed. Also, why was this not included in Figure 2?

These figures are lacking in any statistical analysis, or at least it was not reported. The authors cannot make conclusions without proper analysis. For instance, the claim is made that CTLA-4 antibodies are increased when it looks no different from the GM-CSF vaccine that does not include CTLA-4 antigen. Anti-PD-L1 antibodies look increased, but the statistical tests are missing.

Figure 4:

The legend says N of at least 4, and 3 images per group were shown. What was the rationale for choosing three of them to show?

The %change graph is lacking error bars and statistical tests. The data are not able to be interpreted without them. Also visually it looks like the phenotype is primarily from one mouse with a strong response. This should be discussed. Just from the images, the results do not look consistent.

Again in panels B and C the authors discuss increase of cellular infiltrate. I do not think that claim can be made without quantifiable data.

Other notes:

Additionally, the authors should analyze how well the antibody responses correlate with the %change of tumor growth. If this is meant to be a replacement for ICI therapy, it would be beneficial to directly compare vaccine versus ICI therapy, especially since ICI efficacy on its own is an unknown in this model system.

Final analysis:

Judging from the data presented, I do not think the authors' claims are fully supported. Substantial work is needed, and so I am recommending rejection at this time.

Author Response

In this paper, the authors utilize a TAA-induced iCCA cancer model system in rats. They first detect PD-L1 expression and CD8 T-cell infiltration into the tumor microenvironment by IHC. They showed that a DNA vaccine targeting PD-1 lacked immunogenicity and efficacy. They then showed that a DNA vaccine targeting both PD-L1 and CTLA4 was able to elicit some amount of antibody response, the vaccine was able to reduce the tumor burden in at least one rat, and by IHC the authors claim that CD8 T-cell abundance went up and PD-L1 expression decreased with vaccine.

This is an interesting topic in a model with important human health implications. The idea of using a vaccine to elicit antibodies de novo against ICI's instead of giving the antibody as a therapy is very interesting and novel. That being said, I have major reservations that are outlined below.

Introduction:

The introduction is insufficient, lacking essential topics. For instance it lacks the following, which I consider to be important:

Comment 1: Background about the vaccine platform. Why was a DNA vaccine chosen if the end goal was antibody production? Other vaccine modalities are generally superior at eliciting antibody responses.

Authors’ reply:

DNA, proteins/peptides are used for cancer vaccines. In this manuscript, we firstly studied the potential of CTLA4-PDL1 DNA vaccine in iCCA in vivo. In the future, we will investigate the potential of proteins/peptides vaccines in iCCA

Comment 2: Background about the model system, especially what is known about its immunology, its response to ICIs, and how well it parallels the human condition. I was unable to find any papers that have attempted ICI therapy in this model, so it is unknown if the vaccine would be efficacious even if it proves to be immunogenic. Similarly, there isn't much data available on the human cancer and how it reacts to ICI. The authors note that only MSI-high or dMMR varieties are recommended to receive ICI treatment. Also reference 20 that shows modest efficacy is in the supplement of the journal and was not accessible by my institution, so I was unable to confirm that data.

Authors’ reply:

Indeed, the immune system of TAA-induced iCCA is unclear. However, the experiments in this manuscript were provided crucial information in vivo and the information may provide the reference for the development of the DNA vaccine of CTLA4-PDL1 in the future. As suggested by the reviewer, we provided the PDF file of reference 20. Moreover, we have added the paragraph in the background selection in the revised manuscript.

“In previous studies (Carcinogenesis. 2004 Apr;25(4):631-6; Ann Surg. 2004 Jul;240(1):89-94), rat TAA-iCCAs have been compared with human iCCAs. In histologic features, multifocal bile ductular proliferation, histologic atypia and invasive intestinal-type CCA were chronologically determined and the expression of c-Met and c-ErbB-2, EGFR, apomucins, MMPs were detected in TAA-induced iCCAs and human CCAs, suggesting the progression of TAA-induced iCCA can mimic the multi-step model of human CCA.”

Comment 3: A better transition to understanding the overall rationale driving this work. It is unclear why the authors are testing out a vaccine against ICIs instead of testing the efficacy of ICIs themselves. I consider this a major logical flaw that needs to be directly addressed. If ICI's don't work in this model, why would vaccines against ICI's work? If ICI's do work in this model, why would a vaccine be a better idea than using the ICI's themselves? This needs explanation.

Authors’ reply:

Thanks for your comments. Indeed, no efficacy of ICIs was examined in this model. Furthermore, to our best knowledge, no anti-rat ICIs are available currently. Our goal is to develop vaccination and possibly replace the current ICIs treatment in humans. DNA vaccines may have the advantage of long-term durable ICI effects comparing with current ICIs treatment. As we mentioned before, this TAA-induced iCCA model can mimic the multi-step model of human CCA so this model is a good candidate for testing DNA vaccination before entering clinical trials. In addition, the rat is an immune-competent model that is better than the immunocompromised or immunodeficient mouse model for vaccination study. For the above reasons, we start this preclinical study to prove this methodology works in the animal model.

Methods:

Comment 4: It is unclear if the IL-12 sequence is the entire protein used as an immune stimulant or if it is the secretion signal only. Section 2.1 on the whole needs revision for clarity.

Authors’ reply:

hIL2ss represents the signal sequence of IL2, which is composed of 20 amino acids and was cloned at the N-terminal of the coding sequences of the DNA vaccines used in this manuscript. According to a previous study (J Gene Med. 2005 Mar;7(3):354-65.), hIL2ss was incorporated into chimeric protein for enhancing secretion. To avoid any confusion, the function and the reference of IL2 have been added in the revised manuscript

Comment 5: Why are the authors using mouse sequences in a rat model? What is the rationale? Why not use the rat sequences? Is there a benefit for using a non-native sequence? Is that translatable? Similarly, what is the rationale for using the human version of IL-12?

Authors’ reply:

Thanks for your critical comments. The mouse sequences have been demonstrated their immunogenicity in the preclinical mouse study. However, the preclinical study used CT26 mouse colon cancer cell lines to demonstrate the efficacy in the immunocompetent mouse model. We would like to validate these sequences in de novo animal cancer model which is close to the human tumor so such TAA-induced iCCA rat model was used. As more than 90% similarity between rat and mouse, we directly used these available mouse sequences in the rat model, and promisingly, these DNA vaccines induced antibody resulting in tumor inhibition. As the result, we would like to publish the concept that DNA vaccines may work in vivo studies. In the future, we will compare the mouse and rat sequences to see if rat sequences have superior activity than mouse sequences. hIL2ss for enhancing secretion has been demonstrated ex vivo (human cell lines) and in vivo (nude mice; J Gene Med. 2005 Mar;7(3):354-65.) and the 20 amino acid sequences of mIL2ss and ratIL2ss are the same. Thus, we used these sequences in our rat model.

Comment 6: This paper is missing quality control measures of the DNA vaccine construct. Was the stock pure? Was it sequence verified? Did the authors confirm that it does produce the protein of interest after cellular transfection? Is that protein secreted?

Authors’ reply: We had previously conducted these quality control studies. The purity of pVAX1-hIL2ss-mPD1 or pVAC-hIL2ss-mCTLA4-mPDL1 was determined by the OD260/OD280 ratio and the agarose gel electrophoresis and the accuracy of DNA sequences were determined by DNA sequencing. The expression and secretion of proteins from CHO cells transiently expressing CTLA4-PDL1 or PD1 were detected by ELISA. We have shown the related data in the revised manuscript (Supplementary Figure 5).

Comment 7: How old were the rats at baseline?

Authors’ reply: 39-40 week-old SD rats for PDI experiments and 48-49 week-old SD rats for CTLA4-PD-L1 experiments.

Comment 8: It is unclear if these results were repeated or the result of one in vivo experiment each. At least two independent experiments are necessary to validate results.

Authors’ reply: In several papers, the animal studies are intra-experiment replication, not inter-experiment replication. In our study, the data were acquired from at least 4 rats. Thus, we did not repeat the experiments.

Comment 9: Statistical methods are mentioned here, but never mentioned that I saw in the figures or text of the results. This is a glaring flaw

Authors’ reply: As suggested by the reviewer, we have described the statistical method in corresponding figure legends.

Results:

Comment 10: section 3.3/Fig 1: If the authors make claims that cells are increased, then there must be quantifiable data presented. IHC pictures are valuable, but they must be corroborated with quantifiable data to make claims about increases of cellular infiltrates, otherwise, the results may be unintentionally biased. To help with the issue of bias, the analysis should also be done in blinded fashion.

Authors’ reply: As suggested by the reviewer, we have added the quantified data of IHC images, and the data are shown in Figures 1B and 1C in the revised manuscript.

Figure 2/3:

Comment 11: Figure 3 has an additional group as compared to figure 2; mGM-CSF-mEGF. This group was not sufficiently introduced and it is unclear what purpose this group serves. I'm assuming it is a control for a vaccine that induces inflammation, but it is not discussed. Also, why was this not included in Figure 2?

Authors’ reply: GM-CSF (Granulocyte-macrophage colony-stimulating factor), a glycoprotein cytokine, secreted by mononuclear leukocytes. GM-CSF induces several effects on the immune system, including dendritic cell maturation, macrophage activation, neutrophil proliferation, and T cell activation. GM-CSF stimulates the antigen-presenting cells and enhances overall immunity disproportionally against various endogenous antigens (J Immunother Cancer. 2014 May 13;2:11; Blood. 2008 Jan 15;111(2):485-91; Blood. 2009 Aug 13;114(7):1289-98). EGF activates EGFR-mediated signals to promote tumor progression in cholangiocarcinoma (J Hepatol. 2014 Aug;61(2):325-32.). The antibody of EGF was produced in the rats receiving EGF proteins to influence the EGF-mediated EGFR signals, preventing tumor progression. In our experiments, mGM-CSF-mEGF protein acted as a positive control for the suppression of tumorigenesis in rats. If it is feasible, I would like to explore the possibility to formulate this manuscript focusing on immune checkpoint vaccine, while leaving the data of GM-CSF-EGF for another manuscript.

Comment 12: These figures are lacking in any statistical analysis, or at least it was not reported. The authors cannot make conclusions without proper analysis. For instance, the claim is made that CTLA-4 antibodies are increased when it looks no different from the GM-CSF vaccine that does not include CTLA-4 antigen. Anti-PD-L1 antibodies look increased, but the statistical tests are missing.

Authors’ reply: As suggested by the reviewer, we have added the statistical tests in the revised manuscript. About CTLA-4 antibodies in the rats receiving GM-CSF, it could be that GM-CSF stimulates the antigen-presenting cells and enhance overall immunity disproportionally against various endogenous antigens, such as PD-L1 and CTLA-4 in our data. The enhanced titer of antibody against CTLA-4 by GM-CSF-EGF was less prominent as CTLA-4-PD-L1 vaccine, suggesting the specific effect of CTLA-4-PD-L1 DNA vaccine on humoral immunity against CTLA-4. In our future study, it would be more informative if different antigens both related and unrelated to immune checkpoints could be evaluated at a longer time point after immunization.

Figure 4:

Comment 13: The legend says N of at least 4, and 3 images per group were shown. What was the rationale for choosing three of them to show?

Authors’ reply: We showed the “representative” images in this experiment. As suggested by the reviewer, we have shown the images from 4 rats in the revised manuscript.

Comment 14: The %change graph is lacking error bars and statistical tests. The data are not able to be interpreted without them. Also visually it looks like the phenotype is primarily from one mouse with a strong response. This should be discussed. Just from the images, the results do not look consistent.

Authors’ reply: As suggested by the reviewer, we have added the error bars and statistical tests in the revised manuscript. In clinical, some patients receiving ICIs can decrease tumor growth and some patients can prevent the cancer worsening. In our experiments, the tumors were sustainable development in the control rats and the tumors in the rats receiving CTLA4-PD-L1 were suppressed or stopped growth, suggesting CTLA4-PD-L1 can reduce tumor iCCA growth in rats.

Comment 15: Again in panels B and C the authors discuss increase of cellular infiltrate. I do not think that claim can be made without quantifiable data.

Authors’ reply: As suggested by the reviewer, we have added the quantified data of IHC images, and the data are shown in Figures 4B and 4C in the revised manuscript.

Reviewer 3 Report

Dear Authors,

Thank you for submitting your work to Vaccines.

This is a very interesting study, examining the potential of ICI vaccines n CCa management.

The experimental model that you suggest is well described and sound, thus the lab work of the study is commendable.

The Introduction provides adeqate background information on the topic. 

There are a couple of points I would like to highlight.

The first one is with regards to statistics. You have used a mix of parametric and non parametric statistics, both descriptive and comparative. It is necessary to identify the profile of your data and focus on the appropriate methodology. Due to the sample size of your study I anticipate that your data do not demonstrate normal distribution. I would thus suggest presenting your data with non parametric descriptors, such as median and interquartile range. Consequently, comparisons should be examined using non parametric tests, such as MWU, as you have done, and Independent Samples Median Test (for comparing medians accross more than two groups).

The second point would be that a paragraph describing the shortcomings of the study. For instance, how does biological behaviour of rat TAA-ICC compare with human ICC? What endpoints of your study could be replicated in clinical studies?

I would thus suggest the above revisions, and I would be happy to reassess the revised manuscript.

KR

Author Response

Thank you for submitting your work to Vaccines.

This is a very interesting study, examining the potential of ICI vaccines n CCa management.

The experimental model that you suggest is well described and sound, thus the lab work of the study is commendable.

The Introduction provides adequate background information on the topic. 

There are a couple of points I would like to highlight.

Comment 1: The first one is with regards to statistics. You have used a mix of parametric and non parametric statistics, both descriptive and comparative. It is necessary to identify the profile of your data and focus on the appropriate methodology. Due to the sample size of your study I anticipate that your data do not demonstrate normal distribution. I would thus suggest presenting your data with non parametric descriptors, such as median and interquartile range. Consequently, comparisons should be examined using non parametric tests, such as MWU, as you have done, and Independent Samples Median Test (for comparing medians across more than two groups).

Authors’ reply: As suggested by the reviewer, we have added the quantified data and analyzed the data using Mann-Whitney U-test in the revised manuscript.

Comment 2: The second point would be that a paragraph describing the shortcomings of the study. For instance, how does the biological behavior of rat TAA-ICC compare with human ICC? What endpoints of your study could be replicated in clinical studies?

Authors’ reply:

As suggested by the reviewer, we have added the paragraph in the discussion selection in the revised manuscript.

“In previous studies (Carcinogenesis. 2004 Apr;25(4):631-6; Ann Surg. 2004 Jul;240(1):89-94), rat TAA-ICCs have been compared to human ICCs. In histologic features, multifocal bile ductular proliferation with intestinal metaplasia and increased histologic atypia were observed in SD rats were administered with TAA 300 mg/L for 9 weeks. At the 16th week, cytokeratin (CK19)-expressing invasive intestinal-type CCA with stromal desmoplasia was obvious. The invasive CCAs were detected in 100% in TAA-administered SD rats at the 22nd week. In gene expressions, c-Met and c-ErbB-2, EGFR, apomucins, MMPs were strongly detected in TAA-induced iCCAs and human CCAs. The progression of TAA-induced iCCA can mimic the multi-step model of human CCA.”

In the clinical study, we hypothesize that the patients receiving short-term DNA vaccination can generate durable antibody which can be detected for a long period of time. Next, the anti-tumor effects may be shown in clinical trials. Finally, the safety profile should be acceptable.